# Targeting Mediators of Inflammation in Heart Failure: A Short Synthesis of Experimental and Clinical Results

**DOI:** 10.3390/ijms222313053

**Published:** 2021-12-02

**Authors:** Timea Magdolna Szabo, Attila Frigy, Előd Ernő Nagy

**Affiliations:** 1Department of Biochemistry and Environmental Chemistry, George Emil Palade University of Medicine, Pharmacy, Sciences and Technology of Targu Mures, 540142 Targu Mures, Romania; timea.szabo@umfst.ro; 2Department of Cardiology, Clinical County Hospital Mures, 540103 Targu Mures, Romania; attila.frigy@umfst.ro; 3Department of Internal Medicine IV, George Emil Palade University of Medicine, Pharmacy, Sciences and Technology of Targu Mures, 540103 Targu Mures, Romania; 4Laboratory of Medical Analysis, Clinical County Hospital Mures, 540394 Targu Mures, Romania

**Keywords:** inflammation, heart failure, pro-inflammatory cytokines, mitochondria, interleukins

## Abstract

Inflammation has emerged as an important contributor to heart failure (HF) development and progression. Current research data highlight the diversity of immune cells, proteins, and signaling pathways involved in the pathogenesis and perpetuation of heart failure. Chronic inflammation is a major cardiovascular risk factor. Proinflammatory signaling molecules in HF initiate vicious cycles altering mitochondrial function and perturbing calcium homeostasis, therefore affecting myocardial contractility. Specific anti-inflammatory treatment represents a novel approach to prevent and slow HF progression. This review provides an update on the putative roles of inflammatory mediators involved in heart failure (tumor necrosis factor-alpha; interleukin 1, 6, 17, 18, 33) and currently available biological and non-biological therapy options targeting the aforementioned mediators and signaling pathways. We also highlight new treatment approaches based on the latest clinical and experimental research.

## 1. Introduction

Heart failure (HF) is linked to increased mortality, frequent hospitalizations, and poor quality of life, showing a significant increase in age-adjusted prevalence. The characteristic clinical presentation of HF includes dyspnea, fatigue, reduced physical capacity, and edema, which are the consequences of congestion and low cardiac output (causing hypoperfusion), the main pathophysiological features of HF. Although currently available evidence-based treatment has significantly reduced patient morbidity and mortality, HF still remains a complex clinical syndrome difficult to tackle [1,2]. Since it was first hypothesized in 1984, abundant and strong evidence emerged proving that sustained neurohormonal activation may exert deleterious effects and impose serious hemodynamic consequences [3,4]. Updated clinical practice guidelines on HF drug therapy highlight the importance of targeting the sympathetic nervous system, the renin-angiotensin-aldosterone system, the natriuretic peptide system, and the sodium–glucose cotransporter-2 (SGLT2) pathway [1,2]. Despite currently available pharmacological and implantable device therapy, disease progression suggests unexplored underlying pathophysiological mechanisms [5]. In 1990, Levine et al. proposed the involvement of inflammation in the pathogenesis of HF, linking elevated levels of tumor necrosis factor-alpha (TNF-α) to cardiac cachexia [6]. Ever since, accumulating preclinical and clinical data on local and systemic inflammation in the promotion and progression of HF has been witnessed. The relationship between inflammation and HF is bidirectional and intricately entwined—ventricular systolic and diastolic dysfunction enhances local and systemic inflammation, whereas an increased inflammatory response promotes myocardial fibrosis and adverse ventricular remodeling [7,8].

Clonal hematopoiesis of indeterminate potential (CHIP) was recently identified as a risk factor for cardiovascular disease, increasing all-cause mortality. Certain somatic mutations in hematopoietic stem and progenitor cells lead to clonal expansion of abnormal leukocytes exhibiting impaired immune responses [9]. CHIP further emphasizes the importance of inflammation in cardiovascular disease and may provide valuable information regarding inflammation in HF.

Numerous experimental studies and clinical trials targeted various local and systemic inflammatory biomarkers and immune cells for the purposes of reducing systemic inflammation and improving myocardial function but showed inconsistent results. Active involvement of systemic inflammation in the complex pathophysiology of HF is a fundamental reason for targeting inflammatory cytokines; however, the local, organ-specific effects of cytokines are another major issue to be considered. Furthermore, the intricate, mutual interactions between the proinflammatory milieu and mitochondrial dysfunction suggest that the pharmacological targeting of cytokines is a medical necessity. Thus, even if inflammation is both a cause and a consequence of heart failure [8], the interruption of vicious cycles between the above two pathophysiological processes seems to be a promising objective.

We discuss here the implications and challenges of specific anti-inflammatory treatments of HF and the potential positive impact on disease progression and prognosis.

## 2. Systemic Inflammatory Cytokines Influence Intracellular Calcium Homeostasis and Myocardial Contractility

Despite common symptoms, heart failure with reduced ejection fraction (HFrEF) and heart failure with preserved ejection fraction (HFpEF) show different pathophysiological backgrounds [10]. HFrEF is frequently associated with postischemic conditions, toxic cell necrosis, trauma, and valvular disease, and is a consequence of cardiomyocyte damage. In contrast, HFpEF is associated with extracardiac factors, such as obesity, diabetes, chronic obstructive pulmonary disease, hypertension, and/or renal failure. Inflammation is involved in both; however, its role may vary according to the type of HF. For example, sterile local inflammation is primarily characteristic of HFrEF, whereas metabolic risk-induced, low-grade systemic inflammation is a feature of HFpEF [11]. 

Several proinflammatory cytokines and chemokines, such as TNF-α, interleukin 1β (IL-1β), interleukin 6 (IL-6), interleukin 18 (IL-18), or monocyte chemoattractant protein 1 (MCP-1), are intensely synthesized and released in HFpEF, and their plasma levels are proportional with the organ deterioration [10]. These cytokines might originate in cardiac structural cells, such as cardiomyocytes, endothelial cells, or fibroblasts, and from infiltrating inflammatory cells, like macrophages or extracardiac tissues [12]. However, some observations reporting the elevation of TNF-α, IL-1β, and IL-6 have a limited sample size or lack proper adjustment. Some studies found increased concentrations of TNF receptors (TNFR1 and TNFR2) or elevated levels of molecules pertaining to the TNF receptor superfamily, like Fas or osteoprotegerin (OPG) [10,13].

Heart failure is an inherent consequence of structural and functional remodeling. Structural alterations comprise cardiomyocyte hypertrophy, a myofibroblastic transformation of cardiac fibroblasts, along with reactive extracellular fibrosis. This remodeling impairs both systolic and diastolic functions of the heart, also increasing the risk of proarrhythmia. Many of the pathogenic consequences at the tissue level are mediated by angiotensin II, which induces lung fibrosis and low-level lung clearance through ERK1/2 kinases. Angiotensin II also stimulates mitogen-activated protein kinase (MAPK), triggering myocardial hypertrophy, and upregulates cyclooxygenase 2 (COX-2) in cardiac fibroblasts [14].

Further, in vivo studies showed that angiotensin II enhanced cardiac and renal synthesis of IL-6 and TNF-α [15]. Increased expression of endothelin 1 and downregulation of natriuretic peptides maintain proinflammatory cytokine secretion, being associated with enhanced production of nitric oxide (NO) by inducible nitric oxide synthase (iNOS), which depresses myocardial contractility [16]. Systemic proinflammatory cytokines generate flagrant specific effects on the myocardium in animal studies: infusion of TNF-α and IL-1β impairs left ventricular contractility and increases the operating end-diastolic volume several hours after administration. In addition, TNF-α and IL-6 modulate the myocardial calcium current and Na^+^/Ca^2+^ exchange, reducing peak intracellular Ca^2+^ levels. Additionally, TNF-α diminishes myofilament calcium sensitivity [17]. IL-1β, IL-6, and TNF-α all possess immediate and delayed cardiodepressant properties. They act on constitutive nitric oxide synthase (cNOS), producing baseline levels of NO. In animal studies, they stimulate iNOS quickly (30 min), further raising NO levels and generating reactive oxygen species (ROS), especially peroxynitrite [17]. NO-dependent loss of β-adrenergic receptor responsiveness is another mechanism involved in reducing contractility. Animal studies proved that mitochondrial calcium is a key modulator of energy metabolism and regulates mitochondria-induced cell death. However, some ambiguities still persist regarding the association between mitochondrial failure, calcium starvation or overload [18].

Mitochondria in cardiomyocytes are highly adaptable organelles that generate adenosine triphosphate (ATP), crucial for contractility and functional cell integrity. On the other hand, ATP released from apoptotic cells represents another danger signal, causing a noticeable change in some cytokine expression profiles: IL-1β, IL-6, chemokine C-X-C motif ligand 1 (CXCL-1), and MCP-1 are all upregulated [8].

Proinflammatory cytokines indirectly tune calcium homeostasis, as IL-1β and TNF-α signaling suppress the Ca^2+^ recycling effector, the sarcoplasmic reticulum Ca^2+^ ATPase (SERCA2), via nuclear factor kappa-light-chain-enhancer of activated B cells (NF-κB) [19]. High blood pressure triggers the NOD-like receptor (NLR) family pyrin domain containing 3 (NLRP3) inflammasome in cardiomyocytes through calcium/calmodulin-dependent protein kinase II-delta (CaMKIId) activation. Consecutively, macrophage activation and infiltration lead to fibrosis, compromising contractility [20]. Calcium leakage via the ryanodine receptor type 2 (RYR2) mutant channels or the influx provoked by BCL2/adenovirus E1B interacting protein 3 (BNIP3) overexpression contributes to cardiomyocyte apoptosis and, finally, to heart failure [21]. 

Two defensive mechanisms, mitophagy and the mitochondrial unfolded protein response, counteract mitochondrial deterioration. Urolithin A, a mitophagy inducer, diminishes systemic inflammation mediated cardiac depression, improving mitochondrial function [22]. In mice, deletion of the regulatory FUN14 domain-containing protein 1 (FUNDC1) inhibits calcium influx from the cytoplasm through the inositol 1,4,5-trisphosphate receptor type 2 (IP_3_R_2_) and maintains mitochondrial integrity.

Mitochondrial performance can be improved by targeted pharmacological treatment. Specific anti-inflammatory agents showed efficacy in improving mitochondrial function. Etanercept, a soluble p75 TNFR fusion protein, protected the activity of respiratory chain complexes III and V, reduced cardiac apoptosis, and partially improved left ventricular end-diastolic volume and ejection fraction when administered subcutaneously twice weekly in a 0.5 mg/kg dose to mongrel dogs subjected to intracardiac pacing [23]. Elamipretide, a peptide derivative drug, improved mitochondrial function by enhancing oxygen flux in complexes I and II, increasing the activity of complexes I and IV and facilitating the recruitment of complex IV to its supercomplex. The drug did not affect normal hearts and did not modify cardiolipin composition [24]. 

## 3. Mitochondrial Danger Signals Trigger Cardiomyocyte Inflammasomes

Mitochondria are double membrane-bound organelles and represent almost one-third of the volume of cardiomyocytes [25]. The tricarboxylic acid cycle generates reducing equivalents, NADH + H^+^ and FADH2, which then enter the chain reaction of oxidative phosphorylation and are re-oxidized, transferring their protons and electrons to oxygen, generating water, while ATP is produced. During the electron transfer, protons are pumped out in the intermembrane space, creating membrane potential, which is then kept during the lifespan of mitochondria. The imperfect reduction of oxygen results in ROS. Thus, mitochondria, besides functioning as a natural power plant, regulate many different essential pathways: the redox state of cells, cell growth, survival and apoptosis, epigenetics, and inflammation [18].

Mitochondrial dysfunction in HF is associated with the release of ROS, N-formyl peptides, cardiolipin, and mitochondrial DNA (mtDNA), which all function as danger signals [26]. MtDNA is vulnerable, lacking protective histones and being located close to the source of free radicals. The hypomethylated islets of mtDNA resemble bacterial sequences and function as damage-associated molecular patterns (DAMPs), inducing the activation of macrophages and generating arthritis in experimental animals. Further, oxidized mtDNA has overt proinflammatory effects as a single, oxidatively transformed base in injected oligonucleotides, also triggering disease [27]. Nucleotide instability in the mtDNA occurs especially when transcription factors are unbound or damaged by oxidative stress. The primary intracellular sensors of mtDNA are representatives of Toll-like receptors (TLR), mainly TLR9, NLRP3, and cyclic GMP-AMP synthase (cGAS). TLR9 is abundant in endolysosomes and senses viral and bacterial DNA and oligonucleotides with unmethylated CpG motifs [28]. TLR9 and NLRP3 activation induce the caspase-1-mediated pro-IL-1β release, and TLR9 also activates IL-6 and TNF-α through NF-κB. These cytokines recruit circulating inflammatory cells, endocytose the secreted mtDNA from the extracellular compartment, and exhibit the same TLR9 and NF-κB activation signature [26,29]. However, the inhibition of mtDNA release with cyclosporine A did not bring significant results [26].

NLRP3 inflammasome activation, besides TLR signaling, is another critical step in maintaining chronic low-grade inflammation by increasing levels of proinflammatory cytokines and leading to cardiomyocyte apoptosis and hypertrophy [30,31]. The role of autophagy in inflammasome clearance has been described in different chronic inflammatory diseases [31]. Proteins of the mitochondrial associated membranes, like phosphofurin acidic cluster sorting protein 2 (PACS2) and autophagy-related protein 14 (ATG14), work together to form the autophagy-initiating complex; therefore, their downregulation suppresses mitophagy [32], while molecules essential for endothelial cell glycosylation and development, like Nogo-B-interacting protein (NgBR), modulate the enrichment of mitochondria-associated endoplasmic reticulum membranes (MAMs), and downregulate the proliferation of smooth vascular cells through suppression of pAkt-IP3R3 signal transduction [33]. NRLP3 knockout mice show an overt improvement in cardiovascular function and metabolic benefits—better glucose tolerance and significantly less atherosclerotic plaques when fed a cholesterol-rich diet [34]. 

Binding of the third sensor molecule, cGAS, to mtDNA determines further complexation with STING, an endoplasmic reticulum-anchored cytosolic protein, activates NF-κB through TRAF family member-associated NF-κB activator-binding kinase (TANK), and initiates a type I interferon response [35].

## 4. A Central Molecule of Proinflammatory Cytokine Signaling, NF-κB, Interferes with Inflammasomes, Perpetuating Mitochondrial Deterioration

NF-κB has long been considered an ancient, inducible transcription factor that converts early IL-1β, TNF-α, and TLR signals into proinflammatory and cell survival gene signatures. There are two distinct pathways involving two different forms of NF-κB: the canonical and the alternative signaling [36,37]. The canonical pathway is triggered by IL-1, TNF-α, or TLR ligand binding. The IκB kinase complex (IKK) phosphorylates the IκB kinase in a site-specific manner, forming the RelA/p50 complex (NF-κB I), which activates proinflammatory cytokines. RelA can modulate these pathways in different tissues; while RelA is vital for activating pro-inflammatory macrophages after lipopolysaccharide treatment, its presence is not necessary for the inflammatory response of hematopoietic cells.

Further, the deletion of some subunits of IKK revealed a cytoprotective role of NF-κB, describing its involvement not only in the onset but also in the resolution of the inflammatory response [36]. The alternative pathway is more characteristic of the innate immune response. It is activated by lymphotoxin B, CD40 ligand, receptor activator of nuclear factor-kappa-Β ligand (RANKL), being involved primarily in the processing of the precursor molecule, p100, instead of the phosphorylation-induced activation and degradation of IκB, resulting in the formation of the RelB/p52 complex (NF-κB II), which is then translocated into the nucleus. The targets of NF-κB proteins comprise genes of key functional proteins, among which are many proinflammatory cytokines and chemokines, along with their transcription factors [37,38].

Several proinflammatory cytokines promote the formation of the IKK complex, thus setting up selfamplification loops for NF-κB. IL-18 augments cytokine expression in an autocrine manner via IKK in neutrophils, NF-κB, and RelA phosphorylation [39]. Serin/threonine kinase 24 is a direct enhancer of interleukin-17A- (IL-17A) mediated signaling, which interacts with transforming growth factor-β-activated kinase 1 (TAK1) and IKKβ, and promotes the TAK1/IKK complex formation. This results in the upregulation of IKKβ/NF-κB with consequent downstream activation of IL-6, TNF-α, CXCL-2, and C-C motif chemokine ligand 20 (CCL20) [40]. IL-17A stimulation triggers IKK, and deletion of IKK abolishes the downstream Act1-TRAF2/5 complex formation, but not the signaling via Act1-TRAF6-NF-κB [41]. Another cytokine, interleukin 15 (IL-15), interacts in cerebral endothelial cells, where it induces phosphorylation and degradation of IκB and phosphorylation and nuclear translocation of the p65 subunit of NF-κB [42]. Interleukin 33 (IL-33) triggers the synthesis of IL-6 (via p65 of the NF-κB) and TNF-α (via the p38–MAPK-activated protein kinases 2/3 pathway) in rodent dendritic cells [43].

It is essential that NF-κB stimulated by proinflammatory cytokines mediates further inflammasome activation, functioning as a transcriptional activator for NLRP3 and pro-IL-1β genes. Both genes contain NF-κB binding sites in their promoter region [37,44]. NF-κB stimulation is linked mechanistically to reduced mitochondrial respiration. Pharmacological or genetic inhibition of the IκB/NF-κB signaling exerts overt changes in mitochondrial protein structure and function. Experimental evidence shows that NF-κB suppression ameliorates respiratory function, attenuating the loss of vital mitochondrial proteins, such as succinate dehydrogenase or adenine nucleotide translocase (ANT) type 1, and enhances mitophagy at the same time [45].

There are also metabolic interferences of the low-grade sterile inflammation with a putative primary role in HFpEF, as glucose and palmitic acid overload induce canonical NF-κB signalization coupled with mitochondrial dysfunction. Suppression of IκBα and NF-κB reduced ROS production even in nutrient overload [45]. In tumor cells, NF-κB can bind to ANT, consequently diminishing ATP synthesis [46]; moreover, it inhibits the peroxisome proliferator-activated receptor-γ (PPAR-γ) coactivator 1α (PGC-1α), an essential regulator of mitochondrial biogenesis and respiration, increasing the production of ROS. Inhibition of NF-κB also improves insulin resistance even in nutrient excess [45].

A summary of the inflammatory pathways, the involvement of NF-κB, and their influence on cardiomyocyte function are represented in Figure 1.

## 5. Targeting Macrophages as a Primary Cytokine Source and Macrophage-Derived Chemokines in Heart Failure

The human immune system encompasses a collection of interconnected immune cells and mediator proteins that play pleiotropic roles integrating adaptive and maladaptive biological processes [47]. The concept of cardioimmunology emphasizes the major contribution of the immune system to the development, structure, and function of the heart [48]. Cardiac tissue consists of cardiomyocytes and cardiac nonmyocyte cells. The latter group mainly includes endothelial cells, fibroblasts, and leukocytes. The most abundant cardiac cells in healthy human adults are endothelial cells, whereas myocytes and leukocytes account for 31.2 ± 5.6% and 2.8 ± 1.2%, respectively. About 80% of murine cardiac leukocytes are myeloid CD11b+ cells [49]. After immunostaining, two subsets of human cardiac macrophages were identified in ischemic and nonischemic HF: CCR2+HLA-DRhigh and CCR2−HLA-DRhigh cells. CCR2+ and CCR2− cardiac macrophages express different surface markers and show different origin, function, and localization patterns. Fetal, tissue-resident, anti-inflammatory CCR2− cells were identified in close proximity to endothelial cells, whereas peripheral monocyte-derived CCR2+ macrophages were located near scar tissue and facilitated the expression of numerous inflammatory proteins (chemokine receptors, chemokines, NF-κB, and other transcription factors, cytokines, such as IL-1β, IL-6), growth factors associated with left ventricular remodeling, as well as extracellular matrix-degrading proteases (MMP9), and tissue inhibitors of metalloproteinases (TIMP1) [50]. These findings were consistent with results obtained in a streptozocin-induced and db/db diabetic cardiomyopathy experimental model, where CCR2 inhibition and CCR2^−/−^ mice showed improved cardiac function. CCR2 was also necessary for inflammatory M1 macrophage cumulation in murine hearts affected by diabetes. The absence of CCR2 in the cardiac tissue of diabetic mice stimulated macrophage polarization toward a reparative phenotype (M2) [51]. CCR2-modulating therapy already showed promising results in preclinical studies. Pharmacological targeting of CCR2 in a rodent myocardial infarction model decreased the area of infarction and reduced the number of M1 inflammatory cells [52]. Inhibition of murine CCR+ monocytes and macrophages in early stages of pressure overload decreased left ventricular adverse remodeling and systolic dysfunction [53]. Several ongoing clinical trials targeted CCR+ cells in chronic low-grade inflammatory diseases such as type 2 diabetes mellitus, HIV infection, cancer, and atherosclerosis. CCR2 modulation could exert anti-inflammatory and, therefore, cardioprotective effects in human HF.

## 6. Targeting Inflammatory Cytokines in Heart Failure

### 6.1. TNF-α

TNF-α is one of the most extensively studied pro-inflammatory biomarkers in HF [54]. The overexpression of TNF-α in the failing myocardium is a known fact [55]. Two TNF-α receptors were described in the adult human heart: TNFR1 and TNFR2 [56]. TNF-α exerts different modulating effects in a receptor-dependent manner. In a myocardial infarction murine model, the TNFR1^−/−^ mice showed improved left ventricular contractility and decreased transcript levels of cytokines such as IL-1β, IL-6, TGF-β, and chemotactic protein-1, whereas TNFR2^−/−^ rodents showed an upregulation of TNFR1 and transcript levels of IL-6 and IL-1β in noninfarcted myocardium, accompanied by adverse remodeling and an exacerbation of left ventricular systolic dysfunction [57]. Exposing the myocardium to large amounts of TNF-α resulted in a negative inotropic effect and induced left ventricular dilation, which were reversible upon the limitation of the exposure [58]. 

Two more extensive clinical trials targeted TNF-α in HF. The RENEWAL (Randomized Etanercept Worldwide Evaluation) analysis of the RECOVER and RENAISSANCE trials using etanercept, a soluble TNF-α receptor, to antagonize the harmful effects of TNF-α, failed to meet the primary endpoint addressing all-cause mortality or hospitalization for HF [59]. In the ATTACH (Anti-TNF Therapy Against Congestive Heart failure) study, HF patients were treated with infliximab, a recombinant DNA-derived chimeric human–mouse IgG monoclonal antibody. The inhibition of TNF-α in HF patients using high-dose infliximab (10 mg/kg) was associated with an increased risk for hospitalization and death of any cause [60]. In addition, when exposed to transmembrane TNF-α, infliximab could cause cell lysis in a complement-dependent manner [61]. Similar deleterious effects in the failing heart could impose significant risks. The results of both trials emphasize the dual role of TNF-α in the pathogenesis of HF and require a more selective blockade of TNF-α signaling.

### 6.2. IL-1

The IL-1 family currently encompasses 11 cytokines with proinflammatory and anti-inflammatory properties [62]. The first two members of the IL-1 family—IL-1α and IL-1β—have been extensively studied in cardiovascular disease [63]. The binding of IL-1α and IL-1β to their interleukin-1 receptor (IL-1R1) culminates in NF-κB activation and precursor pro-IL-1β production. The assembly of the NLRP3-inflammasome is promoted by pathogen-associated molecular patterns (PAMPs) and DAMPs. Pro-IL-1β is subsequently activated via the effector caspase-1 of the NLRP3-inflammasome, resulting in active, proinflammatory IL-1β [64]. Repeated intraperitoneal administrations of recombinant murine IL-1β caused a transient suppression in left ventricular systolic function in prior healthy mice [65]. Furthermore, gevokizumab, an anti-IL-1β antibody, reduced adverse ventricular remodeling in a rat model of ischemic heart failure [66]. Increased circulating levels of IL-1β are strong predictors for long-term outcomes in idiopathic dilated cardiomyopathy [67]. In the CANTOS (Canakinumab Anti-Inflammatory Thrombosis Outcomes Study) trial, patients with prior myocardial infarction and elevated high-sensitivity C-reactive protein (hs-CRP) levels, treated with 150 mg canakinumab, a human anti-IL-1β monoclonal antibody, every 3 months during followup, showed a 15% relative risk reduction in the primary composite endpoint of death, nonfatal stroke, and nonfatal myocardial infarction compared to patients receiving standard treatment [68]. A pre-specified subanalysis of the CANTOS trial aimed to further prove the beneficial effects of canakinumab in ischemic HF patients with hs-CRP concentrations ≥ 2 mg/L: canakinumab reduced hospitalization from heart failure and the composite of hospitalization from heart failure or heart-failure-related mortality in a dose-dependent manner [69]. A post hoc analysis of two clinical trials on anakinra, a recombinant human IL-1 receptor antagonist, in systolic HF patients, showed significant improvement in left ventricular ejection fraction compared to a placebo [70]. Patients with HF and acquired hematopoietic TET2 mutations responded better to canakinumab treatment than patients without TET2 variants [71].

### 6.3. IL-6

IL-6 is part of the IL-6 family of cytokines that use the same glycoprotein 130 kDa (gp130) receptor subunit required for intracellular signaling mediated by Janus kinase 1 (JAK1) and signal transducers and activators of transcription 3 (STAT3) [72]. Elevated circulating levels of IL-6 in HF were associated with higher plasma N-terminal pro-brain natriuretic peptide (NT pro-BNP) and renin concentrations and increased risk of mortality (adjusted HR = 2.3, 95% CI = 1.5–3.7, *p* < 0.001) [73]. In HF, elevated levels of circulating IL-6 were associated with highly differentiated T-lymphocyte subsets—CD4+ and CD8+ cells, showing a more accelerated immunosenescence [74]. The relationship between the differentiation of T-cells and proinflammatory cytokines is bidirectional: differentiated T-cell phenotypes increase serum concentrations of proinflammatory proteins, whereas low-grade inflammation is involved in the aging of T-lymphocytes [75]. Although IL-6 is described as a proinflammatory cytokine linked to abnormal ventricular remodeling and systolic function, emerging evidence suggests pleiotropic actions of IL-6 [76]. Myocardial IL-6 and gp130 mRNA showed no significant increase in end-stage heart failure compared to donor hearts [77]. One possible explanation may rely on the fact that IL-6 is also involved in trans-signaling via soluble interleukin-6 receptors (sIL-6R). This mechanism allows IL-6 signaling in cells without surface IL-6R [72]. Tocilizumab (TCZ), an IL-6R neutralizing humanized monoclonal antibody, is currently approved for the treatment of Castleman disease, rheumatoid arthritis, systemic and polyarticular juvenile idiopathic arthritis, giant cell arteritis, Takayasu arteritis, and cytokine releasing syndrome [78]. Seventy patients with active rheumatoid arthritis and unknown cardiovascular disease treated with TCZ at a dose of 8 mg/kg every 4 weeks showed a 63% reduction in circulating levels of NT pro-BNP after 24 weeks of treatment (109.0 pg/mL vs. 39.8 pg/mL, *p* < 0.0001) [79]. In a double-blind, randomized, placebo-controlled phase 2 trial, a single dose of 280 mg TCZ was administered intravenously to patients with non-ST segment elevation myocardial infarction prior to coronary angiography. Patients who received TCZ showed lower levels of hs-CRP compared to the placebo group (4.2 vs. 2.0 mg/L/h, *p* < 0.001) [80]. Further research needs to be conducted to assess the safety of IL-6 blockade in HF.

### 6.4. IL-17

The proinflammatory IL-17A is the hallmark of the six-membered IL-17 cytokine family (A–F) [81]. Although CD4+ T helper type 17 (Th17) cells are the major source of IL-17A [82], IL-17A is also secreted by neutrophils, natural killer (NK) T cells, mast cells, gamma delta T cells, group 3 innate lymphoid cells, Tc17 (CD8+) cells, and microglia [81]. IL-17A is overexpressed in chronic and acute inflammation in both autoimmune and pathogen- (mainly fungi and bacteria) induced inflammatory diseases [81]. Gut dysbiosis was also frequently linked to abnormal IL-17A-mediated immune response [83]. PAMPs and stress proteins activate antigen-presenting cells (APCs) and stimulate IL-1β and interleukin 23 (IL-23) production [84]. IL-23 further induces Th17 cell population differentiation and IL-17A expression [85]. IL-17 also proved to play an essential role in chronic cardiac allograft rejection. T-bet transcription factor knockout mice showed an increased expression of IL-17, IL-6, and interleukin 12p40 (IL-12p40) accompanied by early accelerated diffuse coronary artery intimal hyperplasia and CD4+ Th17 cell infiltration [86]. IL-17^−/−^ cardiac allograft recipients temporarily depleted of CD4+ T cells showed reduced interstitial fibrosis promoted via TGF-β induced IL-17-mediated cardiac fibroblast proliferation and connective tissue deposition [87]. IL-17 also promotes contractile dysfunction by decreasing the expression of the pore-forming subunit Cav1.2 of the L-type calcium channel and SERCA2a, therefore reducing the calcium current in cardiomyocytes. The regulatory effects of IL-17 are mediated via the NF-κB signaling pathway by increasing the expression of p50 [88]. IL-17 concentrations increase with advanced HF as reflected by higher NT pro-BNP levels and worsening NYHA functional class [89]. Elevated IL-17 levels were accompanied by a corresponding increase in CD4+ Th17 cell count in end-stage congestive HF (NYHA class IV) compared to the control group and patients with NYHA class I HF [90]. Three anti-IL-17 agents are currently available and recommended to treat psoriasis in patients with concomitant congestive HF [91]. Secukinumab and ixekizumab directly bind to IL-17A, whereas brodalumab occupies the IL-17 receptor A, preventing IL-17A from activating the receptor—all antagonizing the IL-17A signaling pathway [92]. In the meta-analysis by Champs et al. on cytokine-neutralizing antibodies in psoriatic arthritis and psoriasis, anti-IL-17 treatment with secukinumab, ixekizumab, and brodalumab showed no significant short-term risks of major adverse cardiovascular events or congestive HF when compared to a placebo [93]. IL-17A may therefore represent a potential therapeutic target in the management of HF.

### 6.5. IL-18

IL-18 belongs to the IL-1 superfamily of cytokines among IL-1α and IL-1β. IL-18 is a pleiotropic proinflammatory cytokine synthesized as pro-IL-18, which requires prior activation by caspase-1, an intracellular cysteine protease [62]. Pro-IL-18 differentiates itself from pro-IL-1β mainly by its widespread cellular expression in healthy individuals [62]. The effects of IL-18 are neutralized by the endogenous IL-18 binding protein (IL-18BP), which binds circulating IL-18 and prevents the cytokine from activating its receptors (IL-18Rα and IL-18Rβ), inhibiting interferon gamma (IFN-γ) release from Th1 and NK cells [62]. IL-18 promotes myocardial fibrosis and remodeling. Fibroblasts harvested from adult male rat hearts showed increased expression of extracellular matrix components in a dose-dependent manner once exposed to increasing concentrations of IL-18 (1–100 ng/mL). The highest IL-18 dose (100 ng/mL) significantly enhanced fibroblast migration and proliferation [94]. Male mice receiving plasma from HF patients and murine IL-1β showed reduced left ventricular fractional shortening, a consequence prevented by prior administration of recombinant human IL-18BP and IL-18 blocking antibodies. IL-1β failed to impair left ventricular systolic function in IL-18 and IL-18 receptor knockout mice. Toldo et al. also proved that IL-1β induces cardiac dysfunction via IL-18 signaling [95]. IL-18 represents an independent predictor of poor prognosis in HF patients aged 60 or older [96]. The IL-18/IL-18Rα/IL-18BP expression is significantly altered in the failing human myocardium. Cardiac IL-18 mRNA levels were increased in patients with ischemic cardiomyopathy. The protective effects of IL-18BP were lost in the failing heart—myocardial IL-18BP mRNA expression was significantly downregulated in both ischemic (0.25 ± 0.05, *p* < 0.0001) and dilatative cardiomyopathy (0.18 ± 0.02, *p* < 0.0001) [97]. Ridker et al. showed substantial residual inflammation related to IL-18, IL-6, and associated cardiovascular risk, after IL-1β inhibition with canakinumab in the CANTOS trial [98]. Recombinant human IL-18BP was administered at a dose of 2 mg/kg subcutaneously every 48 h in a white female patient under age 1 with de novo heterozygous NLRC4 mutation, elevated levels of IL-18, and resistance to corticosteroid, IL-1 blockade (anakinra), TNF-α blockade (infliximab), cyclosporine, and α4β7-integrin inhibitor (vedolizumab) treatment [99]. Currently available scientific evidence urges approval of anti-IL-18 therapy in humans and the development of pharmacological agents that simultaneously inhibit both IL-1β and IL-18.

### 6.6. IL-33

The IL-1 family has several members, including the more recently discovered IL-33, a nuclear cytokine predominantly expressed in endothelial, epithelial, and fibroblast-like cells. It acts on a series of immune cells by activating the transmembrane ST2 receptor (also known as ST2L), mainly targeting regulatory T cells, mast cells, and group 2 innate lymphoid cells. ST2 then binds to the IL-1 receptor accessory protein (IL-1RAcP) and promotes NF-κB activation, similar to IL-1β and IL-18. Unlike IL-1β and IL-18, IL-33 does not require prior maturation by caspase-1. Its precursor, IL-33 full-length (IL-33FL), also possesses biological activity. IL-33FL is cleaved by inflammatory proteases, resulting in mature, more active IL-33. IL-33 proved to have both proinflammatory and anti-inflammatory properties [100]. IL-33 protects against atherosclerosis: the number of atherosclerotic plaques was significantly lower in the IL-33 treated ApoE^−/−^ mice compared to the control, phosphate-buffered saline-treated group; additionally, mice receiving soluble ST2 (sST2), an IL-33 neutralizing decoy receptor, developed more extensive aortic sinus atherosclerosis compared to ApoE^−/−^ mice solely injected with the control IgG [101]. Patients with advanced nonischemic HF (NYHA III and IV) participating in the Prospective Randomized Amlodipine Survival Evaluation 2 (PRAISE-2) study had significantly higher levels of circulating ST2 compared to subjects with unimpaired left ventricular systolic function (median 0.24 ng/mL versus 0.14 ng/mL, *p* < 0.0001). A 2-week increase in sST2 levels proved to be an independent predictor of death and heart transplantation (adjusted OR = 1.56, 95% CI = 1.11–2.47, *p* = 0.029) [102]. Circulating ST2 levels were also predictors of mortality and heart failure in patients suffering from acute myocardial infarction [103]. IL-33 exerts antihypertrophic effects and reduces left ventricular remodeling induced by angiotensin II and phenylephrine [104]. Higher myocardial concentrations of IL-33, sST2, and IL-1β were detected in rats subjected to acute myocardial infarction, compared to the littermates without permanent left anterior descending artery ligation. Additionally, both anakinra and eplerenone decreased circulating ST2 levels [105]. Further research on IL-33/ST2 signaling needs to clarify the exact role played by IL-33 in LV remodeling in order to identify novel pathway modulators, which could selectively increase cardiac IL-33 and/or reduce sST2 and prevent maladaptive LV hypertrophy. 

Anticytokine treatment is a promising approach in antagonizing the harmful effects of systemic inflammation. Although currently available results are equivocal, the majority of evidence indicates mainly beneficial or neutral outcomes. This further highlights the dual role played by inflammation in the pathogenesis of HF, and also raises challenges in implementing future research designs (Table 1).

## 7. Targeting NF-κB

Invoking observations that NF-κB expression is upregulated in human myocardial tissue in the heart failure of various etiologies, in parallel with transcriptionally regulated cytokines (IL-1, TNF-α), adhesion molecules (ICAM-1, VCAM-1), or effector enzymes (iNOS, COX-2), the possibility of specific NF-κB inhibition treatment was proposed two decades ago [106]. In early studies, pimobendan, a veterinary drug that suppresses NF-κB, proved to be helpful in the management of heart failure, reducing the number of cardiovascular events and improving quality of life (EPOCH study) [107]. In animal ischemia models, the IκBα super-repressor improved ventricular remodeling, systolic function, and survival [108]. In mice, cardiac-specific deletion of RelA reduced hypertrophy in response to pressure overload [31]. However, several NF-κB inhibitors, acting on inducer kinases, the proteasome, or IκBα, showed high toxicity [109]. Currently, the most promising NF-κB inhibitor targets the family members of G protein-coupled Receptor Kinases (GRK) GRK2, GRK5, and GRK6. GRK2 and GRK5 both phosphorylate IκBα, releasing NF-κB for nuclear translocation. A peptide resembling the RH domain of GRK5 (TAT-RH) interacts with the formation of the IκB/NF-κB complex and DNA binding, decreases cardiac mass, and prevents hypertrophy in rats [109]. A synthetic peptide inhibitor of GRK2 designed as the catalytic domain of GRK2 linked to the antennapedia internalization sequence (Ant124) prevented left ventricular hypertrophy and inhibited NF-κB expression in an experimental model [110].

## 8. Targeting Gut Dysbiosis in Heart Failure

Gut microbiome dysbiosis represents a qualitative and quantitative alteration in the intestinal flora associated with numerous cardiovascular and noncardiovascular diseases frequently linked to low-grade inflammation, including HF [111,112]. Pathogenic gut bacteria are more common in patients with the advanced New York Heart Association (NYHA) functional class [113]. A proposed and plausible explanation implies intestinal epithelial barrier dysfunction and increased permeability due to systemic venous congestion, intestinal mucosal ischemia, and gastrointestinal dysmotility [113,114]. Gut ischemia in HF develops because of low cardiac output and intestinal vasoconstriction mediated by increased sympathetic activity. Enhanced intestinal permeability facilitates the translocation of bacterial cell components from the intestinal lumen into the blood [114]. The outer membrane of Gram-negative bacteria found in the gastrointestinal microbiota contains lipopolysaccharides (LPSs), consisting of a hydrophobic domain better known as lipid A or endotoxin, a nonrepeating oligosaccharide, and a distal polysaccharide chain [115]. Circulating intestinal LPS triggers a systemic inflammatory response mediated via TLR4 and NF-κB signaling pathways, increasing circulating levels of TNF-α, IL-1, and IL-6 [111]. Pattern recognition receptors (PRRs), especially TLRs located in the cell membrane of macrophages, dendritic cells, fibroblasts, and epithelial cells, may be the primary targets of excessive bacterial LPS, triggering the release of proinflammatory cytokines and low-grade tissue inflammation. PRRs can recognize pathogen-specific signatures, PAMPs, such as LPS. 10 different TLRs were identified in humans (TLR1-10) [116]. Expression of TLR4 is upregulated in the failing heart [117]. Chronic activation of TLR signaling promotes adverse cardiac remodeling and increases circulating levels of proinflammatory cytokines [118]. In a murine model of pressure-overload HF, inhibition of TLR4 using eritoran, a synthetic lipid A antagonist, reduced cardiac hypertrophy by decreasing IL-1β and IL-6 levels and increasing anti-inflammatory IL-10 concentrations [119]. Treating female rats with pharmacologically induced heart failure with prebiotics improved intestinal dysbiosis [120]. Diuretic treatment reduces systemic venous congestion, correcting altered gut–blood barrier permeability [114]. Beyond its known lipid-lowering properties, fluvastatin also reduced TLR4 expression in monocytes of HF patients [121]. Soluble CD14 (sCD14) released by LPS-activated monocytes and macrophages proved to be a reliable marker of increased intestinal permeability [122]. Blood drawn from 20 patients with HF was treated with IC14, an anti-CD14 antibody, which reduced circulating TNF-α levels in response to prior LPS (*E. coli*, serotype 0111:B4) stimulation [123].

## 9. Targeting Systemic Inflammation by Means of Exercise Training

Exercise intolerance is defined as the decreased ability or inability to perform physical activity. Reduced exercise capacity is frequently encountered in HF, leading to impaired quality of life and poor prognosis. Several factors, such as cardiac reserve, lung function, peripheral vascular reserve, skeletal muscle dysfunction, obesity, anemia, contribute to exercise intolerance in HF patients [124]. Currently available HF guidelines firmly recommend using regular exercise for patients with HF to improve quality of life, functional capacity, and to reduce the number of hospital admissions from HF [1,2]. Taylor et al. recently updated their previous meta-analysis on exercised-based rehabilitation in HF, including 44 studies and a total of 5783 patients. Exercise-based cardiac rehabilitation improved quality of life assessed using the Minnesota Living with Heart Failure questionnaire and reduced all-cause hospitalization (RR: 0.70; 95% CI: 0.60 to 0.83) and hospitalization from HF (RR: 0.59; 95% CI: 0.42 to 0.84) [125,126]. Emerging evidence suggests that regular exercise mainly exerts antioxidant and anti-inflammatory effects by targeting the cardiovascular system, skeletal muscles, adipose tissue, and immune system [127]. IL-6 is the first myokine released into the bloodstream by skeletal muscle cells upon muscle contraction. Regular exercise maintains elevated levels of serum IL-6. Although IL-6 is primarily linked to proinflammatory states, elevated circulating levels of muscle-derived IL-6 are not accompanied by a corresponding increase in serum TNF-α and IL-1β, thus exerting rather anti-inflammatory effects [128]. Exercise inhibits, through gene methylation, an adaptor protein of the NLRP3 inflammasome activation and an apoptosis-associated speck-like protein containing a CARD (ASC), as well as decreases IL-1β expression [30]. IL-6 also increases serum levels of anti-inflammatory markers, such as IL-1 receptor antagonist (IL-1ra) and IL-10 [129]. IL-10 exerts an inhibitory effect on TNF-α and IL-1β production, further reducing inflammation [130]. Additionally, regular exercise reduces adipose tissue, particularly visceral fat, which contains trapped macrophages within, and represents a significant source of TNF-α [131,132]. Exercise reduced the number of monocyte-derived inflammatory M1 macrophages in the adipose tissue of obese mice and stimulated macrophage polarization toward a reparative phenotype—M2 [133]. Although proven effective, exercise training and cardiac rehabilitation are not commonly used [134].

## 10. Conclusions

Currently available scientific evidence highlights the major role played by inflammation in the pathogenesis and progression of HF. This has to be considered in designing and introducing new, innovative therapies. At the same time, further research needs to be conducted in order to provide missing information on linking signaling pathways and explanations as to why immunomodulating therapy failed to reduce mortality.

## Figures and Tables

**Figure 1 ijms-22-13053-f001:**
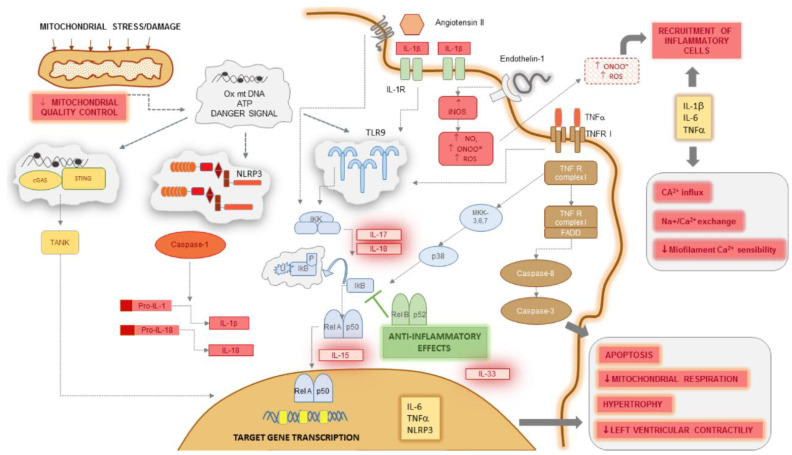
Inflammatory pathways trigger cardiomyocyte dysfunction in heart failure. Mitochondrial damage due to excessive mechanical and oxidative stress and decrease in mitochondrial quality control releases oxidized and hypomethylated mtDNA fragments and ATP, which along with other end-products, represent danger signals inside cardiomyocytes. These are perceived by three sensor systems: TLR9, the NLRP3 inflammasome, and the cGAS/STING complex. TLR9 triggers the IKK kinase that inactivates IκB, an inhibitory molecule on the Rel A/p50 complex of NF-κB; consecutively, this translocates to the nucleus. IL-6, TNF-α, and NLRP3 are among the primary target genes transcriptionally regulated by NF-κB. NLRP3 activates caspase-1, generating elevated levels of IL-1β and IL-18. TLR9 receives other activating signals from ligand-binding IL-1 and TNF RI complexes. The latter may also induce caspase-8 and caspase-3 through FADD, sensitizing the cell to apoptosis. IL-1β, IL-6, and TNF-α induce hypertrophy and suppress contractility, perturbing Ca^2+^ homeostasis and reducing myofilament Ca^2+^ sensitivity. When secreted in the extracellular space, they also recruit inflammatory cells together with peroxynitrite radicals generated by excessive activation of iNOS. Several other proinflammatory cytokines interact with critical steps of NF-κB activation, as IL-17 and IL-18 trigger IκB, IL-15 facilitates its nuclear translocation, whereas IL-33 upregulates gene transcription. For these reasons, IL-1β, IL-6, TNF-α, IL-17, IL-18, and IL-33 may all be putative therapeutic targets in heart failure.

**Table 1 ijms-22-13053-t001:** Clinical trials targeting systemic inflammation—HF-related results.

Author	Study	Drug	Treatment Duration	No. of Patients	NYHA Class LVEF	Endpoint(s)	Outcome Reached?
*TNF-α blockade*							
Mann et al. [59]	RENEWAL: RECOVER RENAISSANCE	Etanercept	24 weeks	1500 1123 925	II-IV ≤30%	Reduction in the risk of the composite end point of all-cause mortality or HHF. Improvement in clinical status	No
Chung et al. [60]	ATTACH	Infliximab	14 weeks	150	III-IV ≤35%	Improvement in clinical status	No
Champs et al. [93]	Meta-analysis of RCTs in PsA or psoriasis	Adalimumab	12–24 weeks	1250	Not specified	Increased risk of MACE or CHF in patients with PsA or psoriasis treated with adalimumab	No
Champs et al. [93]	Meta-analysis of RCTs in PsA or psoriasis	Etanercept	12–24 weeks	1398	Not specified	Increased risk of MACE or CHF in patients with PsA or psoriasis treated with etanercept	No
Champs et al. [93]	Meta-analysis of RCTs in PsA or psoriasis	Infliximab	10–30 weeks	696	Not specified	Increased risk of MACE or CHF in patients with PsA or psoriasis treated with infliximab	No
*IL-1β blockade*							
Ridker et al. [68]	CANTOS	Canakinumab	Median of 3.7 years	10,061	Not specified	Reduction in the risk of the composite end point of CV death, stroke, or MI	Yes
Everett et al. [69]	CANTOS Subanalysis—HHF	Canakinumab	Median of 3.7 years	10,061	Not specified	Reduction in the risk of the composite end point of HHF or HF related mortality	Yes
Svensson et al. [71]	CANTOS subanalysis—CHIP due to TET2	Canakinumab	Median of 3.7 years	3925	Not specified	Increased risk of MACE in the TET2 mutation subgroup	Yes
Buckley et al. [70]	Post-hoc study of 2 CTs	Anakinra	2 weeks	80	II-III <40%	Increased LV systolic performance	Yes
*IL-6 blockade*							
Yokoe et al. [79]	Effects of TCZ on NT pro-BNP in RA	Tocilizumab	24 weeks	70	No CVD	Decrease in NT pro-BNP levels	Yes
Kleveland et al. [80]	Effects of TCZ on inflammation in NSTEMI	Tocilizumab	Single dose	117	Not specified >50%	Reduction in hs-CRP and hs-TnT release in NSTEMI	Yes
*IL-17 blockade*							
Champs et al. [93]	Meta-analysis of RCTs in PsA or psoriasis	Brodalumab	12 weeks	834	Not specified	Increased risk of MACE or CHF in patients with PsA or psoriasis treated with brodalumab	No
Champs et al. [93]	Meta-analysis of RCTs in PsA or psoriasis	Ixekizumab	12–24 weeks	776	Not specified	Increased risk of MACE or CHF in patients with PsA or psoriasis treated with ixekizumab	No
Champs et al. [93]	Meta-analysis of RCTs in PsA or psoriasis	Secukinumab	12–36 weeks	1343	Not specified	Increased risk of MACE or CHF in patients with PsA or psoriasis treated with secukinumab	No

HF, heart failure; NYHA, New York Heart Association; LVEF, left ventricular ejection fraction; HHF, hospitalization for heart failure; RCTs, randomized control trials; PsA, psoriatic arthritis; MACE, major adverse cardiac events; CHF, congestive heart failure; CV, cardiovascular; MI, myocardial infarction; CHIP, clonal hematopoiesis of indeterminate potential; CTs, clinical trials; LV, left ventricular; TCZ, tocilizumab; NT pro-BNP, N-terminal pro-brain natriuretic peptide; RA, rheumatoid arthritis; CVD, cardiovascular disease; NSTEMI, non-ST-elevation myocardial infarction; hs-CRP, high-sensitivity C-reactive protein; hs-TnT, high-sensitivity cardiac troponin T.

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
