# Peer review of "Targeting Mediators of Inflammation in Heart Failure: A Short Synthesis of Experimental and Clinical Results"

_ijms, 2021, doi:10.3390/ijms222313053_

Round 1

Reviewer 1 Report

The authors conducted a review article highlighting the role of chronic inflammation on Cardiovascular disease with specific focus on HFrEF and inflammation mediated perturbations in mitochondrial function and calcium homeostasis. They highlight current therapeutic strategies targeting inflammatory mediators and signaling pathways in HF.

Here are my suggestions:

  • In the Abstract section, line 25, I suggest changing “new treatment possibilities” to “New treatment approaches or avenues”.
  • The phrase: furthermore, the intricate…, line 63-65, is not clear the way it is written. My recommendation would be: Furthermore, the intricate, mutual interactions between the pro-inflammatory milieu and mitochondrial dysfunction suggest that the pharmacological targeting of cytokines is a medical necessity.
  • Please change vicious circles to vicious cycles throughout the manuscript.
  • Line 67: please add pathophysiological processes so that the sentence reads: …between the above two pathophysiological processes seems to be a …
  • The Phrase: while all three…., line 111-114 is not clear and is too long. I suggest that the sentence is split into two phrases and revised for clarity.
  • Line 154, please change NADH+H+ to either NADH or NADH + H+.
  • In figure 1, I suggest that the authors change: decrease in Mitophagy (highlighted in the schematic drawing) to decrease in mitochondrial quality control. Markers of autophagy and mitophagy are up-regulated in systolic HF, However the efficiency of removal of the damaged organelles by macroautophagy is decreased due to saturation of lysosomes with lipofuscin material, and peroxidation and instability of the lysosomal membranes. Thus, the term mitochondrial quality control would be more appropriate.

Author Response

Answer to Reviewer 1

Thank you for your careful and constructive comments. Please find below our point-to point answers to the issues raised.

Reviewer 1

The authors conducted a review article highlighting the role of chronic inflammation on Cardiovascular disease with specific focus on HFrEF and inflammation mediated perturbations in mitochondrial function and calcium homeostasis. They highlight current therapeutic strategies targeting inflammatory mediators and signaling pathways in HF.

 Here are my suggestions: 

  • In the Abstract section, line 25, I suggest changing “new treatment possibilities” to “New treatment approachesor avenues”.
  • The phrase: furthermore, the intricate…, line 63-65, is not clear the way it is written. My recommendation would be: Furthermore, the intricate, mutual interactions between the pro-inflammatory milieu and mitochondrial dysfunction suggest that the pharmacological targeting of cytokines is a medical necessity.
  • Please change vicious circles to vicious cyclesthroughout the manuscript.
  • Line 67: please add pathophysiological processes so that the sentence reads: …between the above two pathophysiological processesseems to be a …
  • The Phrase: while all three…., line 111-114 is not clear and is too long. I suggest that the sentence is split into two phrases and revised for clarity.
  • Line 154, please change NADH+H+ to either NADHor NADH + H+.
  • In figure 1, I suggest that the authors change: decrease in Mitophagy (highlighted in the schematic drawing) to decrease in mitochondrial quality control. Markers of autophagy and mitophagy are up-regulated in systolic HF, However the efficiency of removal of the damaged organelles by macroautophagy is decreased due to saturation of lysosomes with lipofuscin material, and peroxidation and instability of the lysosomal membranes. Thus, the term mitochondrial quality control would be more appropriate.

Authors: we have considered all your suggestions, as follows:

  1. Abstract: “We also highlight new treatment approaches based on the latest clinical and experimental research.”
  2. L62-64: “Furthermore, the intricate, mutual interactions between the pro-inflammatory milieu and mitochondrial dysfunction suggest that the pharmacological targeting of cytokines is a medical necessity.”
  3. We changed vicious circles to vicious cycles:

Abstract: “Chronic inflammation is a major cardiovascular risk factor. Pro-inflammatory signaling molecules in HF initiate vicious cycles altering mitochondrial function and perturbing calcium homeostasis”.

L65-67: “Thus, even if inflammation is both a cause and a consequence of heart failure [8], the interruption of vicious cycles between the above two pathophysiological processes seems to be a promising objective.”

  1. We rephrased the indicated text:

L112-115: “They act on constitutive nitric oxide synthase (cNOS), producing baseline levels of NO. In animal studies, they stimulate iNOS quickly (30 minutes), further raising NO levels and generating reactive oxygen species (ROS), especially peroxynitrite.”

  1. We performed the change indicated:

L154: “The tricarboxylic acid cycle generates reducing equivalents, NADH + H+ and FADH2”.

  1. In Figure 1, we changed decreased mitophagy to decreased mitochondrial quality control. Please see Figure 1 and L257.

Please, see the revised manuscript.

The authors

Reviewer 2 Report

This is a well written and conducted review.

According to my opinion the  authors should change the title by omitting the phrase  ‘mitocondrial background’. This is because they discuss in a limited way the role of mitochondria. Therefore, the title of the manuscript could be ‘Targeting Mediators of  Inflammation in Heart Failure; a Short Synthesis of  Experimental and Clinical Results’.

Additionally, they could provide a Table with anti-inflammatory therapies that they failed to improve heart failure end point. This might reinforce the concept that inflammation and heart failure interaction is not simple as it is presented in several other reports.

Author Response

Answer to Reviewer 2

Thank you for your careful and constructive comments. Please find below our point-to point answers to the issues raised.

Reviewer 2:

According to my opinion the authors should change the title by omitting the phrase “mitocondrial background”. This is because they discuss in a limited way the role of mitochondria. Therefore, the title of the manuscript could be ‘Targeting Mediators of Inflammation in Heart Failure; a Short Synthesis of Experimental and Clinical Results’.

Authors:

According to your opinion, we changed the title which now is: “Targeting Mediators of Inflammation in Heart Failure: A Short Synthesis of Experimental and Clinical Results”.

Reviewer 2:

Additionally, they could provide a Table with anti-inflammatory therapies that they failed to improve heart failure end point. This might reinforce the concept that inflammation and heart failure interaction is not simple as it is presented in several other reports.

Authors:

In the revised version, we provide Table 1, entitled “Clinical trials targeting systemic inflammation – HF-related results,” presenting the most important clinical studies, the applied drugs, endpoints, and outcomes.

Please, see the revised manuscript.

The authors